# Automatic Markerless Motion Detector Method against Traditional Digitisation for 3-Dimensional Movement Kinematic Analysis of Ball Kicking in Soccer Field Context

**DOI:** 10.3390/ijerph19031179

**Published:** 2022-01-21

**Authors:** Luiz H. Palucci Vieira, Paulo R. P. Santiago, Allan Pinto, Rodrigo Aquino, Ricardo da S. Torres, Fabio A. Barbieri

**Affiliations:** 1Human Movement Research Laboratory (MOVI-LAB), Graduate Program in Movement Sciences, Department of Physical Education, Faculty of Sciences, São Paulo State University (Unesp), Bauru 17033-360, SP, Brazil; fabio.barbieri@unesp.br; 2LaBioCoM Biomechanics and Motor Control Laboratory, EEFERP School of Physical Education and Sport of Ribeirão Preto, USP University of São Paulo, Campus Ribeirão Preto, Ribeirão Preto 14040-907, SP, Brazil; paulosantiago@usp.br (P.R.P.S.); aquino.rlq@gmail.com (R.A.); 3Reasoning for Complex Data Laboratory (RECOD Lab), Institute of Computing, University of Campinas, Campinas 13083-852, SP, Brazil; allan.pinto@ic.unicamp.br; 4FMRP Faculty of Medicine at Ribeirão Preto, University of São Paulo, Ribeirão Preto 14049-900, SP, Brazil; 5LabSport, Department of Sports, CEFD Center of Physical Education and Sports, UFES Federal University of Espírito Santo, Vitória 29075-910, ES, Brazil; 6Department of ICT and Natural Sciences, NTNU–Norwegian University of Science and Technology, 6009 Ålesund, Norway; ricardo.torres@ntnu.no

**Keywords:** image processing, human estimation, COCO, MPII, deep learning, team sports

## Abstract

Kicking is a fundamental skill in soccer that often contributes to match outcomes. Lower limb movement features (e.g., joint position and velocity) are determinants of kick performance. However, obtaining kicking kinematics under field conditions generally requires time-consuming manual tracking. The current study aimed to compare a contemporary markerless automatic motion estimation algorithm (OpenPose) with manual digitisation (DVIDEOW software) in obtaining on-field kicking kinematic parameters. An experimental dataset of under-17 players from all outfield positions was used. Kick attempts were performed in an official pitch against a goalkeeper. Four digital video cameras were used to record full-body motion during support and ball contact phases of each kick. Three-dimensional positions of hip, knee, ankle, toe and foot centre-of-mass (CM_foot_) generally showed no significant differences when computed by automatic as compared to manual tracking (whole kicking movement cycle), while only z-coordinates of knee and calcaneus markers at specific points differed between methods. The resulting time-series matrices of positions (r^2^ = 0.94) and velocity signals (r^2^ = 0.68) were largely associated (all *p* < 0.01). The mean absolute error of OpenPose motion tracking was 3.49 cm for determining positions (ranging from 2.78 cm (CM_foot_) to 4.13 cm (dominant hip)) and 1.29 m/s for calculating joint velocity (0.95 m/s (knee) to 1.50 m/s (non-dominant hip)) as compared to reference measures by manual digitisation. Angular range-of-motion showed significant correlations between methods for the ankle (r = 0.59, *p* < 0.01, *large*) and knee joint displacements (r = 0.84, *p* < 0.001, *very large*) but not in the hip (r = 0.04, *p* = 0.85, *unclear*). Markerless motion tracking (OpenPose) can help to successfully obtain some lower limb position, velocity, and joint angular outputs during kicks performed in a naturally occurring environment.

## 1. Introduction

Kicking is a frequent technical action taken in football matches, and achieving high-velocity and accurate targeting standards usually contributes to winning matches [1,2]. Extracting players’ movement kinematics data derived from ball kicking, such as passing actions or shooting against goalposts, is arguably important in the coaching process since it assists in identifying and understanding (a) body mechanical factor determinants for performance outputs; (b) practice constraints’ (e.g., acute-to-chronic exercise or recovery related) effects on coordination aspects [3]; most importantly, (c) individual weaknesses in need of improvement, thereby allowing the provision of specific feedback to refine movement patterns [4]. Currently, among the tools available to examine kicking movement performance, video analysis originates the most objective and sensitive kinematic metrics, which are not always captured by only visual ratings [5]. In fact, video analysis has been used in some other research fields (e.g., animal displacement detection and analysis), as well as in a range of sports [6,7,8,9,10,11,12,13,14]. However, while ecologically valid assessment protocols using an on-field multiple-camera setup (such as high-speed action sport cameras [15]) have made it possible to record the three-dimensional and open nature of soccer kicks, flexible/reliable and boosting image processing methods are still required to provide useful information to practitioners in due time [3,16,17].

Traditional excess of the manual digitisation necessary in obtaining body landmark position time-series information is one important issue related to video kinematic investigations of kicking [18,19,20], which would render as long as half a year of data processing depending on sample sizes [21]. Indeed, manual tracking generally results in the existence of both inter- and intra-operator variability in labelling [22] and is highly dependent on marker sharpness/contrast in relation to the skin and environment [23,24]. Furthermore, game rules do not always allow players to wear objects [25], implying that kick assessment routines should attempt to respect the realities of competition [3]. The latter can also prevent athlete monitoring with foot-mounted sensors [26], or otherwise attaching any apparatus to the player’s body may interfere with their normal technique [27,28,29]. Thus, publicly available markerless motion estimation algorithms seem to be a potential solution in analysing video data and extracting kick kinematics in a likely less time-consuming, more cost-effective, and non-invasive way. For such purposes, advanced computer vision and deep learning techniques (e.g., convolutional neural network (CNN) mask) [30] have been applied whilst the validity of the various contemporary markerless systems is still lacking consensus in research (for a recent review, see Cronin [31]).

To date, results on the performance of a widespread, state-of-the-art open-source pose estimation tool called ‘OpenPose’ [32] are found for a range of dynamic tasks, including human walking [33], running [34], jumping [35], and ball throwing [36]; this method demonstrated acceptable functioning in two- [33,37] or three-dimensional [34,35,36] plane analysis (e.g., agreement with measures derived from ‘gold-standard’ marker-based kinematic systems). Either of these previous experiments assessing movements pertaining to team sports was notably performed indoors in such controlled environments as a laboratory room. To our knowledge, no studies determined its measurement error under field conditions such as a soccer pitch scenario and considering the inclusion of ballistic (explosive) tasks involving object manipulation using the lower limbs such as in ball kicking. Differently from a throw, during kicking actions, one limb is fixed on the ground while another moves in parallel, generating some blocking between these body segments in the camera’s view, which can influence tracking effectiveness [31]. Notwithstanding, as this method involves the recognition/estimation of key body points in each image (i.e., video frame) separately, the implementation of tracking strategies represents an important attempt to further improve such a tool in motion analysis, despite the fact that it was almost overlooked in the existing works specifically interested in sports tasks. Additionally, non-controllable aspects comprising lighting, possible mutual occlusions and diversified subject/background configurations may often collectively interfere during image processing in team sport real-world contexts [13,29,38]. Therefore, the purpose of the current preliminary study was to compare traditional frame-by-frame manual and modified OpenPose automatic motion tracking methods in determining soccer kicking movement kinematic parameters in a naturally occurring environment.

## 2. Materials and Methods

### 2.1. Data Acquisition

In an official, FIFA-standard, natural grass pitch, four tripod-mounted digital video cameras (240 Hz, 1920 × 1080 pixel; Hero 7/Black Edition, GoPro^®^ GmbH, München, Germany) were distributed perpendicular to each other and 2.5 m laterally to the established kick mark (18 m apart from the midpoint goal line). An experimental dataset (unpublished observations) including six outfield—defenders, midfielders and forwards—U17 academy players was used (16.1 ± 1 years old, 176 ± 6 cm, 71 ± 11 kg). Spherical markers (25 mm diameter) were attached externally on the participant’s greater femoral trochanter and also, on the non-dominant side, the lateral malleolus, calcaneus, and distal phalanx of the fifth metatarsal head (dominant limb). As illustrated in Figure 1A, kick attempts were performed near the entrance of the penalty area (18 m from the midpoint goal line), and participants were instructed to strike the target’s centre (1 × 1 m in both goalpost upper corners) while avoiding a teammate goalkeeper intercepting the ball. A detailed description of the design of the field assessments and the test–retest reliability measures of the task adopted to measure soccer kicking characteristics are both available elsewhere [39]. Following collection, image sequences were transferred to a computer Ubuntu 20.04.2 LTS (Intel i7-8750H (12) @ 2.208 GHz; 16 GB RAM; Intel Corporation, Santa Clara, CA, USA) with GPU (GeForce GTX 1050 Ti; 4096 MB GDDR5; NVIDIA Corporation, Santa Clara, CA, USA), and 30 kicking attempts were randomly selected (https://www.random.org; accessed on 8 July 2021) for further analysis.

### 2.2. Motion Tracking

#### 2.2.1. Manual Digitisation

Manual frame-by-frame tracking of markers was completed (Figure 1D) by a single experienced (>10 years) examiner using the DVIDEOW interface (v.5.0; Laboratory of Instrumentation for Biomechanics & Institute of Computing UNICAMP, Brazil—see Figueroa et al. for more information [40]). This motion tracking method is considered as the reference measurement in the current study. The audio-band feature embedded into the software was employed to synchronise the video cameras before tracking [41]. The image sequences of the same initial and final events and the duration (i.e., number of frames) were inputted in DVIDEOW and OpenPose tracking methods. A common calibration frame containing 49 control points was also defined for both in order to completely cover the kicking area (3.6 × 3.2 × 1.3 m, respectively, in antero-posterior, medio-lateral and vertical directions—Figure 1B). Reconstructed calibration points were compared to the reference values entered by the operator to compute a global error.

#### 2.2.2. Markerless System

OpenPose is a bottom-up deep-learning-based approach designed to estimate human pose and joint angles. In brief, this approach estimates and encodes the locations (Figure 1C) and orientations of the limbs, in addition to the association score between body parts. For this, the method produces confidence maps that encode the location of each body part in the image domain. Then, a set of 2D vector fields of part affinity fields (PAFs) is used to encode both location and information. Next, the PAFs are used to estimate the degree of association between the body parts, which is used to assemble the limbs and thus come up with a full-body pose for all people in the scene. A more detailed description of this method can be found in [32]. Although OpenPose can detect the pose of all people in a scene, a tracking strategy is necessary for associating a set of poses found in the frame t with their respective poses detected in a subsequent frame t + 1. In this work, we proposed a simple tracking strategy that firstly consists of computing the centre of mass of a given pose in the frame t. Then, we associate it with the pose in the frame t + 1 whose centre of mass presents the smallest Euclidean distance. We take advantage of the K-Means algorithm [42] for efficiently computing the pairwise distances between a set of poses found in the frame t and frame t + 1. The source code is freely available for scientific purposes on GitHub repository (https://github.com/allansp84/allansp84-markerless-motion-detection, accessed on 9 November 2021).

### 2.3. Data Treatment and Measures

In a Matlab^®^ environment (R2019a; MathWorks Inc., Natick, MA, USA), 3-dimensional coordinates were obtained from Direct Linear Transformation (DLT) [43], with previous correction for radial lens distortion [44] caused by the type of action that the sports cameras used. The raw data from both methods were treated with a dual filter fourth-order Butterworth plus rloess [45,46], using a span equal to 0.1 and a cut-off frequency of 25 Hz as smoothing parameters, which were set after a residual analysis. The dependent variables of the current study were determined for all kick attempts using custom-built routines and included: (a) time-series reconstructed 3-dimensional position of each marker; (b) its associated time-series velocity; (c) foot centre-of-mass (CM_foot_) velocity at impact; (d) angular range of motion (ROM). Velocity outputs were taken from finite central differences using time-series data of the entire attempt duration as input arguments [18]. CM_foot_ velocity at impact was calculated from the three final frames just before ball contact considering the centroid of malleolus, calcaneus, and toe coordinates. Local reference frames and segment centres were defined as per previous studies [21,47]. Joint ROMs were then computed as the difference between the lowest and the highest joint (hip, knee, and ankle) angle values observed in the sagittal plane [47]. Independent variables were the two distinct methods used (DVIDEOW software for manual tracking and OpenPose markerless system) for the digitisation of the kicking motion.

### 2.4. Statistical Analysis

The data were time normalised and presented as medians and 95% confidence intervals (CI). Significant differences between methods in time-series analysis were flagged when CIs did not overlap [21,45,48]. Mean absolute error (MAE) was computed according to Equation (1) [36] separately to support phase (1% to 65% of the cycle—starting from the take-off of the kicking limb until complete touchdown of the supporting limb in the last step before impact), contact phase (from 66% to 100% of the cycle, starting immediately following termination of the support phase until foot–ball impact [45]) as well as considering the whole kick cycle. Correlation coefficients were determined (time-series (corrcoef.m Matlab function) and single-point measures (Pearson product-moment (r) and intraclass correlation coefficients—ICC)) to gain insights about the random error and interchangeability of signals obtained among the two methods. For the latter, the statistical significance level was pre-set at *p* ≤ 0.05. The correlation values (r, 95% CI) were deemed as being trivial (<0.1), small (>0.1–0.3), moderate (>0.3–0.5), large (>0.5–0.7), very large (>0.7–0.9) or almost perfect (>0.9–1.0). If the resulting CIs overlapped both small positive and negative values, then the correlation was deemed unclear [49]. Shared variance was considered small (*r*^2^ < 0.30), moderate (*r*^2^ > 0.3–0.5) or large (*r*^2^ > 0.50). Ratio limits of agreement (RLOA) were also computed for the non-time-series parameters (joint ROMs and CM_foot_ velocity at impact) as a measure of systematic bias [50]. Finally, effect sizes (ES) assumed as standardised mean differences (Cohen’s *d*) were calculated [51] for each comparison (Equation (2)) and interpreted as trivial (*d* < 0.2), small (*d* > 0.2–0.59), moderate (*d* ≥ 0.6–1.19) or large (*d* ≥ 1.2).
(1)$$\mathrm{MAE}=\frac{1}{n}\sum_{i=1}^{n}\left|d_{m}\left(i\right)-d_{a}\left(i\right)\right|$$
(2)$$\mathrm{ES}\left(i\right)=\frac{M_{m}\left(i\right)-M_{a}\left(i\right)}{\sqrt{\frac{\sigma_{m}^{2}\left(i\right)+\sigma_{a}^{2}\left(i\right)}{2}\;}}$$
where $d_{m}$ = the data point obtained from manual tracking; $d_{a}$ = the data point obtained from automatic tracking; n = the number of data samples; i = a given frame; $M_{m}$ = the average obtained from manual tracking; $M_{a}$ = the average obtained from automatic tracking; $\sigma_{m}$ = the standard deviation of the values obtained from manual tracking; $\sigma_{a}$ = the standard deviation of values obtained from automatic tracking.

## 3. Results

Figure 2 shows the time series of the absolute position data (x, y and z coordinates) of selected markers obtained from OpenPose automatic and DVIDEOW manual tracking methods, while Figure 3 contains its three-dimensional resultant velocity observed across the whole kicking movement cycle. Separate figures for all markers/axes are provided in Appendix A. Global calibration errors were 0.9, 1.1 and 1.8 cm, respectively, in the x (antero-posterior), y (medio-lateral) and z (vertical) directions.

Positional differences were observed between the two methods only in the z-coordinates of the knee (from 6 to 12%; 20% instant and 90 to 100% of kicking movement cycle—Figure 2F) and calcaneus markers (from 16 to 18%—Appendix A). Hip (dominant and non-dominant), ankle, fifth metatarsal head and CM_foot_ (Figure 2G–I) showed no significant differences in their three-dimensional positions (x, y or z coordinates), computed by automatic as compared to manual tracking methods, during the whole kicking movement cycle. Overall effect sizes (Table 1) were trivial to small for the x coordinates (ES = 0.01 to 0.39 (mean = 0.17)) and y coordinates (ES = 0.06 to 0.32 (mean = 0.14)) and small to large concerning z coordinates (ES = 0.34 to 6.40 (mean = 1.59)).

In reference to the marker’s velocity outputs, the overall differences between methods were trivial (ES ≤ 0.08 (mean = 0.03)) but reached significance in the non-dominant hip (from 15 to 20%, from 37 to 47%, from 58 to 67%, and from 98 to 100%—Figure 3A), dominant hip (from 1 to 2% and 65% instant—Figure 3B), calcaneus (from 68 to 90%—Figure 3D), fifth metatarsal head (from 5 to 7%, 17% instant, from 64 to 68%, and from 97 to 99%—Figure 3E) and CM_foot_ (from 6 to 8%, 71% instant, and from 99 to 100% of kicking movement cycle—Figure 3F). Knee showed two separate differences (48% and 97% instants—Figure 3C), whilst ankle velocity was similar (Appendix A) between OpenPose automatic and DVIDEOW manual tracking methods (ES ≤ 0.07 (trivial)) across the whole kicking cycle (Appendix A).

The overall MAEs of the OpenPose motion tracking method as compared to manual digitisation was 3.49 cm for determining positions and 1.29 m/s for calculating markers’ velocity (all pooled). In particular (Table 2), these values ranged, respectively, from 2.78 cm (CM_foot_) to 4.13 cm (dominant hip) and 0.95 m/s (knee) to 1.50 m/s (non-dominant hip). ROM showed *large-to-very large* correlations between OpenPose automatic and DVIDEOW manual tracking methods, respectively, for the ankle (r = 0.59 (± 0.40), *p* < 0.01; ICC = 0.47, *p* < 0.01; RLOA = 0.655 ×/÷ 1.758; 1.08 [0.92; 1.24] and 1.41 [1.33; 1.49] rad) and knee joint displacements (r = 0.84 (± 0.08), *p* < 0.001; ICC = 0.82, *p* < 0.001; RLOA = 1.072 ×/÷ 0.323; 1.91 [1.83; 1.99] and 1.86 [1.76; 1.96] rad), but it was *unclear* in the hip (r = 0.04 (± 0.15), *p* = 0.85; ICC = 0.01, *p* = 0.48; RLOA = 0.520 ×/÷ 1.940; 0.86 [0.43; 1.28] and 1.14 [1.08; 1.20] rad), while CM_foot_ velocity at impact showed a moderate correlation (r = 0.48 (± 0.23), *p* < 0.01; ICC = 0.47, *p* < 0.01; RLOA = 1.021 ×/÷ 0.046; 21.57 [20.77; 22.36] and 19.70 [19.04; 20.36] m/s, respectively). Finally, large average shared variances were found among methods in their resulting matrix of position (*r*^2^ = 0.94 [0.73; 0.99], all *p* < 0.001) and velocity signals (*r*^2^ = 0.68 [0.46; 0.92], all *p* < 0.01).

## 4. Discussion

The current study aimed to compare the OpenPose automatic motion detector method [32] against traditional frame-by-frame manual digitisation (DVIDEOW software) conducted through a relatively flexible interface [40] in determining on-field soccer kicking movement kinematics. Furthermore, a simple tracking algorithm was proposed to overcome the original limitation of OpenPose that provides a set of full-body poses per frame without any tracking information, allowing us to determine the set of poses for a specific person over time. To the extent of our knowledge, no previous work has investigated the reliability aspects of markerless tracking in ball kicking drills. In the following paragraphs, the strengths and weaknesses relating to the application of such advanced tracking methods are discussed—in particular, within the context of kicking kinematics analysis—and recommendations for possible further improvements are also provided.

Manual tracking systems (marker-based) are one of the most common computer methods adopted in measuring the kinematics of soccer kicks to date, despite their possible impractical characteristics. According to the original results obtained, a markerless algorithm can help to successfully obtain lower limb position and velocity outputs during kick tasks performed in an official outdoor pitch. Critical parameters related to kicking effectiveness, such as those derived from knee and ankle joint kinematics, showed acceptable levels of error when computed by the OpenPose algorithm as compared to traditional tracking (i.e., manual frame-by-frame tracking of kick movement features [18,19,52,53,54]). Conversely, hip motion demonstrated the worst outcomes using this procedure in all of the position, velocity and angular aspects. Notwithstanding, while the accuracy of the x axis and y axis was generally preserved, most sources of uncertainties when using the contemporary tracking tool were likely due to discrepancies identified in the *z*-axis (vertical) coordinates among methods, indicating that advances in the OpenPose motion detector are still necessary, mainly in the hip segment. Regardless, such a markerless system can be helpful for performance-enhancing purposes (training to correct occasional inefficient movements) as well as possible clinical cases (e.g., identifying athletes with potentially excessive plantar flexion amplitude; Tol et al. [55]) owing to the presumable interchangeability of its signals with manual digitisation.

In biomechanical investigations, the validity of the measures is invariably a concern, notably in field testing where non-controllable/unexpected factors usually exist. In a first analysis, the error values found were within the range reported previously by applying OpenPose to other human movements—primarily performed with the lower limbs—which were found to be 2 cm in walking at a comfortable pace [56], about 3–4 cm in walking plus quick jumping conditions [36] and up to 5 cm on average in sprinting bouts [28]. However, variable acquisition frequencies (30 [56], 120 [36] or 240 [35] Hz), number of cameras (one [33], two [34,35,56], five [36] or nine [28]), their positioning (e.g., 1.8 [34], 2.3 [56] or 6.35 m [33] away from the subjects) and recording resolution (640 × 480 [56], 1280 × 720 [35], 1920 × 1080 p [28] or 4K [36]) may constrain such a direct comparison of our results and literature. In general, such previous studies also used experimental setups including cameras with apparently traditional optical systems (e.g., linear field of view) in which no mention was made concerning whether the image sequences were corrected for eventual distortions. Thus, it is plausible to state that action sport cameras were not tested among existing works, which highlights another innovative aspect of the current study. Regardless, these documented results indicate that, similar to that occurring with other biomechanical position monitoring systems [57], uncertainties of OpenPose may be speed dependent. In accordance, we have verified some increases in error values (e.g., average +8.23% and +15.07% in *x*-axis positions and resultant velocities across markers, respectively) from the support to contact phase of kicking (Table 2). Of note, distal endpoint kicking velocity increased from ~33–37 km/h in toe-off to 84–102 km/h at the ball contact instant, as exhibited in Figure 3E. These sudden changes in velocity could alter the image properties (e.g., might add some blurred pixels), thereby influencing motion estimation [31]. Furthermore, CM_foot_ velocity—a key parameter of performance—showed infrequent but significant systematic bias across time moments of the kicking cycle, including ball impact. Conversely, random error appears to be low given the significant correlations between methods, whilst the RLOA of CM_foot_ velocity at impact is also compatible with previous studies measuring kick velocity [58,59]. In this sense, given the habitual distortion caused by foot–ball impact on the kicking kinematics data, excluding the last few frames of the contact phase is not uncommon [3,45,60] and could also solve this issue despite improvements in the precision of foot detection being mainly required in investigations focused on depicting the full ball-impact phase.

Here, the OpenPose method had superior performance in determining antero-posterior/medio-lateral (respectively, *x* and *y* axis) than vertical (*z* axis) positions (Table 1). The same occurred for intermediate-to-distal (knee, ankle, and foot) limb kinematics—position, velocity and angular amplitude—of the contact limb as compared to the proximal joint (hip) involved in the action (Table 2). It should be noted that the resulting velocity was slightly impacted by the worse results in the z coordinates/vertical direction that are seemingly cancelled by acceptable functioning in the remaining pair of axes. The strong similarity of signals between methods (e.g., as indicated by r-squared outcomes and also the waveforms—Figure 2I and Figure 3C) indicates the presence of more systematic rather than random error. This is further illustrated by the significant associations observed from the resulting ROM (knee and ankle) and CM_foot_ velocity outcomes among the two methods. As shown in Figure 3, all important events such as continuous knee acceleration throughout the support phase followed by a reversal to deceleration—around 70% of the cycle—concomitantly with the foot acceleration, which is a typical illustration of proximal–distal energy transference during kicks, were correctly identified. The highest discrepancies between automatic and manual tracking in z coordinates might be partly attributed to a concomitant greater global calibration error magnitude that was observed in this specific axis, meaning that uncertainties in the vertical direction are themselves the poorest. Lower precision in obtaining hip movements is also observed elsewhere in squat and locomotion tasks [34,37]. The hip is generally entirely covered by clothing (shorts/t-shirt) [61], with unconstrained portions that are not well fixed to the body, and this probably causes an undefined/more variable pattern of appearance in image sequences, while this is not the case or might be at least reduced in the remaining equipment (socks and clothes) covering the leg and foot segments. In addition to being wider, by the under-17 age, the hips and thighs are also longer [21] as compared to the lower leg, thereby involving image processing of a greater number of pixels, perhaps increasing the complexity in keypoint detection of the former. In the case of kicking assessed using only lateral cameras, given the common path of players in the approach run and subsequent ball flight, issues in markerless tracking can be also attributed to a longer mutual occlusion of hip joints as compared to the knee and ankle—in particular, the overlapped proximal region of both thighs in the image sequences across a substantial trial duration. Taken collectively, it can make the process to precisely detect and track the hip joint more challenging in some instances. Therefore, hip kinematic outcomes of kicking should be interpreted with caution when using OpenPose. The possible advantages of including a camera behind the player trajectory (i.e., posterior view) in addition to a higher sampling frequency should be tested in future research.

Additional limitations are also recognised to this study, which collectively indicate taking caution in any generalisations made. Firstly, given the practical difficulties in conducting repeated measurements within soccer clubs, the cross-sectional design of the present comparative analysis prevents the calculation of minimal detectable differences. Secondly, re-training based on our own dataset characteristics may contribute to further reducing errors arising from possible algorithm false detections [62]. Thirdly, a relatively small sample size was used here to evaluate the performance of a markerless algorithm, despite this not being uncommon, as some published pilot or validation studies have included, for example, only two [36,56], six [63], or a maximum of nine participants [64]. Fourthly, assessment outcomes may fail to replicate actual soccer match demands, possibly limiting the results to only testing routines. Finally, as soccer kicking performance is age dependent, a priori extrapolation of results to senior players may not be warranted.

## 5. Conclusions

In short, empirical evidence provided by the current experiment indicates that the OpenPose tracking method may be a reliable tool with which to evaluate soccer kicking kinematics in youth soccer under grassy pitch conditions, providing compatible data to those obtained from traditional frame-by-frame manual digitisation. In particular, we were more confident in knee and ankle joint preliminary outcomes, whilst hip detections should still be improved to reduce between-methods differences. Markerless motion tracking systems such as OpenPose seem to be a pertinent option to help fill a critical gap currently existing between research and soccer practice, mainly in the analysis of ball kicking skills, since it reduces the time spent in processing the data dramatically, while its precision in computing position and velocity is generally not compromised. Future work is recommended to further check the validity of the proposed method against microtechnology devices or ‘gold-standard’ marker-based (semi)automatic tracking measures provided by infrared cameras, despite the difficulties in implementing the latter in the “real-world” rather than within a laboratory or indoor artificial turf soccer pitch.

## Figures and Tables

**Figure 1 ijerph-19-01179-f001:**
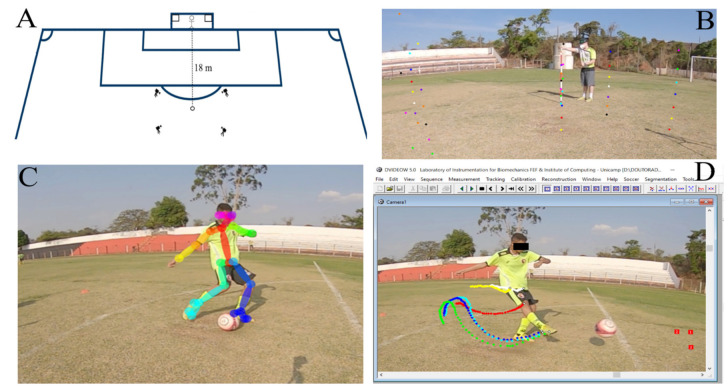
Illustration of (**A**) experimental setup used for data collection, (**B**) calibration, (**C**) a given kicking trial digitised using OpenPose markerless system and (**D**) manual tracking frame by frame.

**Figure 2 ijerph-19-01179-f002:**
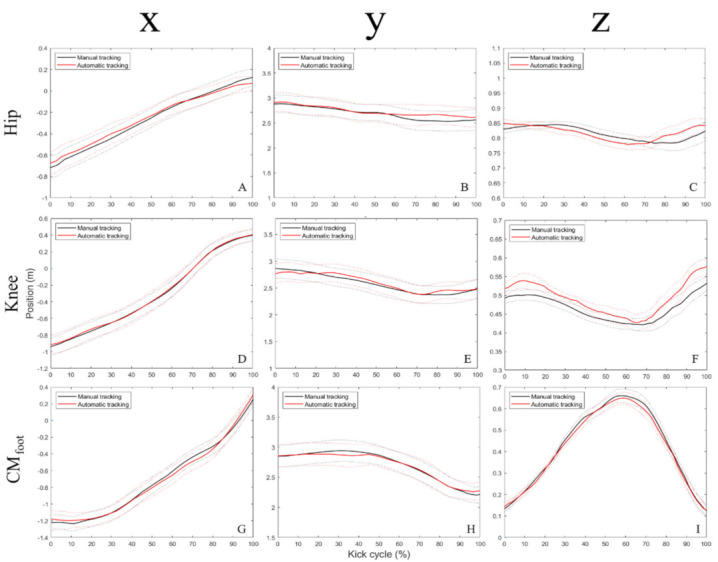
Time series containing median values (solid lines) and associated confidence intervals of marker’s absolute position (x, antero-posterior; y, medio-lateral; z, vertical) computed across the whole kick movement cycle for both tracking methods (dominant hip x-axis (**A**), dominant hip y-axis (**B**), dominant hip z-axis (**C**), knee x-axis (**D**), knee y-axis (**E**), knee z-axis (**F**), CM_foot_ x-axis (**G**), CM_foot_ y-axis (**H**), CM_foot_ z-axis (**I**)).

**Figure 3 ijerph-19-01179-f003:**
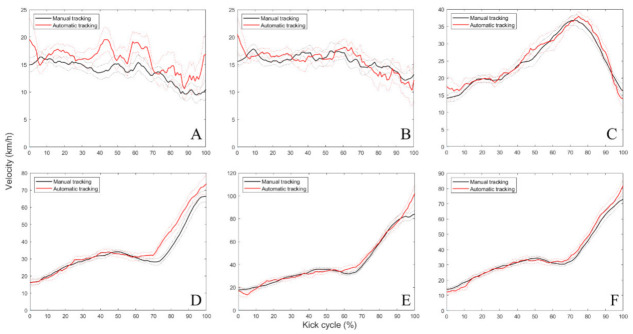
Time series containing median values (solid lines) and associated confidence intervals of marker’s resultant three-dimensional velocity computed across the whole kick movement cycle for both tracking methods (non-dominant hip (**A**), dominant hip (**B**), knee (**C**), calcaneus (**D**), fifth metatarsal head (**E**) and CM_foot_ (**F**)).

**Table 1 ijerph-19-01179-t001:** Effect sizes (ES) obtained for pairwise comparisons between OpenPose automatic tracking and DVIDEOW manual tracking positional data (x, antero-posterior; y, medio-lateral; z, vertical) and velocity outputs.

Marker	Position	Velocity
*X*-Axis	*Y*-Axis	*Z*-Axis
Overall	Min	Max	Overall	Min	Max	Overall	Min	Max	Overall	Min	Max
Non-dominant hip	0.01	−0.14	0.15	−0.10	−0.29	0.11	−0.77	−5.03	4.22	−0.08	−0.25	0.00
Dominant hip	−0.19	−0.34	0.44	−0.32	−0.63	−0.16	1.40	−3.99	6.33	−0.01	−0.11	0.08
Knee	−0.04	−0.39	0.45	−0.15	−0.23	0.00	−6.40	−12.32	−2.78	−0.02	−0.13	0.12
Ankle	−0.39	−0.84	−0.17	−0.15	−0.38	0.05	0.88	−4.71	4.12	0.00	−0.07	0.05
Calcaneus	0.29	−1.07	0.96	0.06	−0.08	0.22	0.75	−4.57	5.50	−0.05	−0.21	0.05
5th metatarsal head	−0.19	−1.71	0.15	−0.10	−0.39	0.05	−0.34	−6.22	1.85	−0.02	−0.11	0.04
CM_foot_	−0.11	−1.23	0.26	−0.07	−0.23	0.10	0.61	−3.76	2.36	−0.02	−0.11	0.05
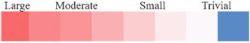

**Table 2 ijerph-19-01179-t002:** Mean absolute error (MAE) for three-dimensional positions (cm) and resultant velocity (m/s) of each joint derived from comparisons between DVIDEOW manual digitisation and OpenPose automatic tracking algorithm.

Marker	Support Phase					Contact Phase					Whole Kick Cycle				
	X	Y	Z	All	VEL	X	Y	Z	All	VEL	X	Y	Z	All	VEL
Non-dominant hip	2.76	1.86	6.54	3.72	1.42	3.11	2.82	7.03	4.32	1.66	2.88	2.20	6.71	3.93	1.50
Dominant hip	2.65	2.49	5.88	3.67	1.06	3.04	2.91	8.98	4.98	1.29	2.79	2.63	6.96	4.13	1.14
Knee	1.91	2.45	4.40	2.92	0.84	1.91	3.19	3.31	2.80	1.14	1.91	2.71	4.02	2.88	0.95
Ankle	3.29	2.09	5.48	3.62	1.54	3.02	1.99	4.68	3.23	1.06	3.19	2.05	5.20	3.48	1.37
Calcaneus	4.44	2.22	5.47	4.04	1.10	4.62	2.59	3.84	3.68	2.12	4.50	2.35	4.90	3.92	1.45
5th metatarsal head	2.60	1.98	5.59	3.39	1.23	3.40	2.13	3.70	3.08	1.81	2.88	2.03	4.93	3.28	1.43
CM_foot_	2.50	1.41	4.87	2.93	1.14	2.93	1.41	3.18	2.51	1.24	2.65	1.41	4.28	2.78	1.17
Overall	2.88	2.07	5.46	3.47	1.19	3.15	2.43	4.96	3.51	1.47	2.97	2.20	5.29	3.49	1.29

VEL = resultant three-dimensional velocity; X = antero-posterior; Y = medio-lateral; Z = vertical directions.

## Data Availability

The raw data supporting the findings of the present study were made available by the authors without undue reservation in the Open Science Framework (https://osf.io/jer9u/, accessed on 14 January 2022) repository.

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
