# Peer review of "Automatic Markerless Motion Detector Method against Traditional Digitisation for 3-Dimensional Movement Kinematic Analysis of Ball Kicking in Soccer Field Context"

_ijerph, 2022, doi:10.3390/ijerph19031179_

Round 1
Reviewer 1 Report
I wish to thank the authors for the opportunity to review their work.
This article presents an interesting study that compares OpenPose video analysis method with manual digitisation tracking method in the evaluation of soccer kicking kinematics under grass pitch conditions in youth soccer.
The research is well executed, and answers interesting questions around markerless motion tracking systems and their usefulness in successfully obtaining lower limbs position, velocity, and joint angular outputs during kicks performed in a naturally occurring environment. The article has some limitations, but they have been correctly identified by the authors, and they have also clearly referenced future lines of investigation.
The article is well written, well structured, the methods are clearly described, and English level is good.
I consider it would be considered relevant by an interested reader, and that it provides new scientific knowledge that could be used in future studies. However, there are some suggestions that could be taken into account to improve the article before its publication:
- L53-54, I would suggest describing briefly (a couple of lines) the use of video analysis in some other fields science, as well as other sports, to emphasize its usefulness. Authors may consider citing some, if not all, of the following references, if they consider them useful:
- Kulyukin, V.; Mukherjee, S. On Video Analysis of Omnidirectional Bee Traffic: Counting Bee Motions with Motion Detection and Image Classification. Sci.2019, 9, 3743. https://doi.org/10.3390/app9183743
- Velázquez, J.S.; Iznaga-Benítez, A.M.; Robau-Porrúa, A.; Sáez-Gutiérrez, F.L.; Cavas, F. New Affordable Method for Measuring Angular Variations Caused by High Heels on the Sagittal Plane of Feet Joints during Gait. Sci.2021, 11, 5605. https://doi.org/10.3390/app11125605
- Amara, S.; Chortane, O.G.; Negra, Y.; Hammami, R.; Khalifa, R.; Chortane, S.G.; van den Tillaar, R. Relationship between Swimming Performance, Biomechanical Variables and the Calculated Predicted 1-RM Push-up in Competitive Swimmers. J. Environ. Res. Public Health2021, 18, 11395. https://doi.org/10.3390/ijerph182111395
- L-193-200 – After reading previous paragraphs, it remains unclear when the kicking cycle starts, and when it ends, i.e., what would be the position assigned to “0%” and which to “100%” in abscises axis in figure 2?. Have the authors considered including a new figure indicating these phases?
- L293-300 – Have the authors considered using a high speed recording camera to reduce blurring of pixels and therefore improve precision of image in last kicking frames? This may help to improve results in Z axis. This could also be cited in line 327.
- Figure 3 – It is difficult to interpret information in this figure, I would recommend to add a brief description in the caption of what is depicted in each image marked with a capital letter (A,B,C, etc.) to improve readability.
Reviewer 2 Report
Dear Authors, the manuscript suggests that there are some important field research findings. However, the written English and grammar style made it difficult to understand the paper.
Reviewer 3 Report
This study investigated the Open-Pose tracking method in the evaluation of the soccer kicking kinematics. Some revisions need to be made as follows:
Introduction
The author should mention the up-to-date markerless motion analysis findings and the potential gaps and the purpose of this study.
Methods
Line 159, the author should explain in more detail about how to determine each segment center of mass and range of motion.
Discussion
Lines 271 – 272, the author needs to explain why hip motion demonstrated worst outcomes.
Lines 288-292, the author mentioned the approach was speed-dependent. While missing the discussion of why speed influences the outcome measures. Would the frame rate also be an influencing factor?
Reviewer 4 Report
The article is interesting and valuable. The work is thematically consistent. The authors say the frame-by-frame studies are accurate. They are looking for a new method of calculating the trajectory by a computer system without reflective markers.The article promotes and develops the subject of the use of optical systems to determine trajectories and other kinematic parameters in sport with the use of FDM.The article presents a new algorithm. The statistical apparatus is at a high level. This article makes a significant contribution to research in a wide variety of sports.The correct way of thinking was presented in relation to the description of the confirmation of the hypotheses related to the test without the use of markers on the sports field The literature review is interesting and the sources are modern. The authors indicated a research gap, but they should clearly refer to the accuracy of the results, taking into account the location of cameras and the optical system of the camera. It was noted by the reviewer that some charts have different qualities. The conclusions are correct. The article is scientifically plausible. The main drawback is the adaptation of the article to the jurnal. A sports journal would be more suitable for presenting the results.
Round 2
Reviewer 2 Report
As the authors have chosen not to further review the English language and style, I am unable to continue with the review.
Reviewer 3 Report
The author has addressed my questions and concerns.
